# Racquet Sports Recognition Using a Hybrid Clustering Model Learned from Integrated Wearable Sensor

**DOI:** 10.3390/s20061638

**Published:** 2020-03-15

**Authors:** Kun Xia, Hanyu Wang, Menghan Xu, Zheng Li, Sheng He, Yusong Tang

**Affiliations:** Department of Electrical Engineering, University of Shanghai for Science and Technology, Shanghai 200093, China; xiakun@usst.edu.cn (K.X.); mhxu12321@163.com (M.X.); lizhengtsyz@126.com (Z.L.); 13817795692@163.com (S.H.); simon.tang@unisoc.com (Y.T.)

**Keywords:** internet of things (IoT), physical activity recognition (PAR), machine learning (ML), wearable sensors

## Abstract

Racquet sports can provide positive benefits for human healthcare. A reliable detection device that can effectively distinguish movement with similar sub-features is therefore needed. In this paper, a racquet sports recognition wristband system and a multilayer hybrid clustering model are proposed to achieve reliable activity recognition and perform number counting. Additionally, a Bluetooth mesh network enables communication between a phone and wristband, and sets-up the connection between multiple devices. This allows users to track their exercise through the phone and share information with other players and referees. Considering the complexity of the classification algorithm and the user-friendliness of the measurement system, the improved multi-layer hybrid clustering model applies three-level K-means clustering to optimize feature extraction and segmentation and then uses the density-based spatial clustering of applications with noise (DBSCAN) algorithm to determine the feature center of different movements. The model can identify unlabeled and noisy data without data calibration and is suitable for smartwatches to recognize multiple racquet sports. The proposed system shows better recognition results and is verified in practical experiments.

## 1. Introduction

Medical research shows that physical exercise can provide positive benefits for human healthcare, including reduced risks of cardiovascular disease, obesity, stroke, and cancer [1], improved musculoskeletal health and stress regulation [2], and reduced psychological health burden and mental disease [3]. Physical activity recognition (PAR), which uses information acquired from a variety of sensors to automatically detect and analyze physical activities [4], has broad applications such as behavior correction and medical detection. PAR can quantify activity levels, improve exercise quality, and reduce healthcare costs. It has been regarded as an important research direction in human–computer interaction. Oja et al. [5] found that racquet sports seem to be the best forms of exercise for reducing the risk of death. Therefore, from the perspective of health and recording exercise effects, it is necessary to provide a reliable racquet sports detection device.

Vision-based PAR mainly uses red-green-blue (RGB) images [6,7,8], optical flow [9], 2D depth maps [10], and 3D skeletons [11,12]. Traditional images are susceptible to illumination variations and camera view angles. Due to inevitable annotation errors, the video dataset is complex for classification. The method based on depth maps and 3D skeletons can provide fine motion recognition, but these methods require expensive special facilities for offline calculations and have a large computational load. Acoustics, vibration, and other environment-based sensors are mostly installed in fixed locations and are not suitable for outdoor activities [13]. The deviceless human activity recognition system mainly uses wireless sensing technology [14]. Researchers use Wi-Fi to obtain channel status information (CSI) or receive signal strength indicators (RSSIs) to classify different activities [15]. This method has limited application scenarios and all prototypes are still in the laboratory test stage [16]. Powerful microcontrollers and energy-efficient wireless data transmission have facilitated the development of wearable technology, which have provided PAR with a less invasive and lower-cost alternative. In a sports science context, inertial sensor data based on accelerometer and gyroscope signals are the most important source of movement analysis [17]. As the most commonly used sensor for acquiring human activity signals, inertial measurement units (IMUs) are widely used in sports recognition [18,19,20,21]. Liu et al. [22] used a body sensor network (BSN) to collect motion data, and a support vector machine (SVM) was used to identify table tennis movements offline. Conaire et al. [23] used a camera and BSN to obtain the contour features and acceleration data of the tennis serve movement. The K-nearest neighbor classifier (KNN) and SVM were used to classify the fused features. Multi-hybrid sensors can improve the accuracy of activity recognition; however, the configuration of multi-hybrid sensors is prone to cause the marker crossover phenomenon and interferes with the user’s normal activities [24]. Racquet sports mainly involve movement of the arm. Although acceleration signals on the trunk can provide better features for basic physical recognition, identifying activities involving the upper limbs is challenging [25]. Considering the user’s acceptance of sensor position and number, it is expected that the movements of racquet sports can be recognized by only a wearable sensor on the wrist.

Similar movements exist in racquet sports, such as service and drive. How to effectively distinguish these movements is a problem. Wang et al. [26] used a two-layer Hidden Markov Model (HMM) to identify 14 types of badminton movements. Fu et al. [27] used convolutional neural network models to identify and analyze the ping-pong movements based on inertial sensing data. However, the above methods cannot provide timely feedback information to users. Wang et al. [28] proposed an intelligent badminton movements recognition system consisting of Bluetooth low energy (BLE) technology, a microelectromechanical systems (MEMS) IMU, cloud technology, and machine learning algorithms. The system only supports access to the recognition results with client Wi-Fi. Despite intensive research, all current devices have their own limitations, mainly classifying the movement of a single-type racket sport, and research on multiple racquet sports movement recognition is lacking in general [29]. A technology that allows instantaneous analysis and data sharing on wearable devices is desired.

The focus of this paper is on an automated stroke detection and classification system of multiple racquet sports. This paper extends existing sensor-based movement measuring methods with a multilayer hybrid clustering model. The proposed model shows good recognition for similar movements in different racquet sports. Meanwhile, a cheap, real-time, and ultra-portable racket sports recognition wristband system is designed to collect data and verify the feasibility of the proposed model.

This paper is organized as follows. Section 2 describes the proposed method, including the wristband system and the multilayer hybrid clustering model. Section 3 validates the system and discusses the experimental results, and the conclusions are presented in Section 4.

## 2. Proposed System

The proposed system includes racquet sports recognition wristbands and smartphones. The architecture of the proposed system framework is presented in Figure 1. Wrist accelerometer and gyroscope signals are collected through the IMU and transmitted to a microprocessor (MCU) by an inter-integrated circuit (I2C) bus. In the training stage, the MCU transmits data to a personal computer (PC) to create datasets, which are used to verify the model designed in this paper. In the exercise, the MCU uses a complementary filter for data preprocessing, extracting features, and analysis based on the multilayer hybrid clustering model, identifies the current movement, performs the number counting, and then displays the information on an organic light-emitting diode (OLED) screen. Data interaction is through a Bluetooth mesh network. The whole process works in real-time.

This section introduces the wristband system in terms of hardware, model, and software. The model, the most important part of this system, is composed of three parts: Data collection, data processing, and a classification algorithm.

### 2.1. Hardware Platform

A new integrated wearable sensor platform has been designed to achieve a miniaturized system (Figure 2). The sensor is equipped with an IMU, MCU, Bluetooth, OLED screen, and battery charge management chip. The size of the sensor is 38 mm × 34 mm × 20 mm. The parameters of the wristband are shown in Table 1. Code is written through a universal serial bus (USB). 

The IMU chosen is the MPU6050, integrating a 3-axis gyroscope and 3-axis accelerometer, and using three 16-bit analog-to-digital converters (ADCs) to convert the measured analog signals into digital signals. The full-scale range of the IMU is programmable, the accelerometer is set to ±4 g, the gyroscope is set to ±2000°/s, and the sampling rate is configured at 50 Hz, which is enough for movement feature collection [30].

The MCU chosen is the STM32F103 series chip in the 48PIN package. This chip uses the ARM Cortex-M3 microcontroller unit. The clock signal is provided by an internal 8 MHz RC oscillator, and the operating frequency is set to 72 MHz, which can provide high-speed online calculations for racket sports models. The MCU integrates timer, control area network (CAN), ADC, serial peripheral interface (SPI), I2C, USB, and universal asynchronous receiver/transmitter (UART) interfaces, which is beneficial to data interaction.

The CC2541 chip is a 2.4 GHz BLE solution and conforms to the Bluetooth v4.0 protocol stack.

The voltage level of the hardware platform is 3.3 V and the regulator chooses the low dropout regulator (LDO) ME6206. To account for actual use scenarios, a low-capacity battery (Li-ion 3.7 V, 500 mAh) has been used. The USB charging circuit chooses the linear Li-ion battery charger TP4056, which uses the P-metal-oxide-semiconductor field-effect transistor (PMOSFET) structure inside and sets an anti-reverse charging circuit to ensure no overcharge.

### 2.2. Data Collection

A total of 5 healthy subjects (3 males, 2 females; age: 25 ± 5) took part in the data collection process. Among them, one subject had received 2 years of professional training in badminton, and one subject had received 4 years of professional training in table tennis. The others were untrained people. All participants provided written informed consent before participation. Subjects were asked to wear the racquet sports recognition wristband on their dominant wrist. Subjects were all right-handed. The datasets were collected in a real training environment (gym).

In the experiment, each subject performed nine kinds of movements: Four types of table tennis (service, stroke, spin, and picking up), four types of badminton (service, drive, smash, and picking up), and walking. Each subject performed 20 tests for each movement. For a subject, the time to complete different movements was different, and the time to complete one movement was between 1 and 1.2 s. The action in the same movement set was collected continuously, and different movement sets were collected separately. Then, 100 instances were collected for each movement set, and a total of 900 instances were collected. During the experiment, the number of actions in each movements set was manually recorded to label the data set at a later stage. 

### 2.3. Data Processing

Preprocessing and feature extraction are needed for the raw data to construct the features that can effectively distinguish racquet sports.

#### 2.3.1. Preprocessing

Median filtering is used to process the noisy raw signals output by MPU6050. The signals of the accelerometer and gyroscope are fused to obtain the angle. As common filtering algorithms, Unscented Kalman Filtering (UKF) [31] and Nonlinear Complementary Filtering (HBL) [14] are considered in the model. Gravitational acceleration (g=9.8) is the benchmark for evaluating filtering algorithms. Combining the acceleration, formulas of HBL (1) and UKF (2) are obtained. The data obtained by the wristband in the static state after filtering with HBL and UKF are compared. The output results are shown in Table 2.
(1)$$ACC_{HBL}=\alpha ACC_{a}+(1-\alpha )ACC_{gyro}.$$

***ACC_HBL_*** is the gravitational acceleration through HBL, where ***ACC_a_*** is the value of the accelerometer, ***ACCgyro*** is the value of the gyroscope, and *α* is the weight coefficient.
(2)$$\{\begin{array}{c} ACC_{k}=ACC_{k-1}\cos(\theta_{k})\cos(\theta_{k})+w_{k}\\ z_{k}=ACC_{k}+v_{k} \end{array}.$$

***ACC_k_*** is the gravitational acceleration through UKF at time k, where ***θ_k_*** is the rotation angle at time k, ***w_k_*** is the process noise at time k, and ***v_k_*** is the measurement noise at time k.

The data obtained by HBL are closer to the theoretical value of gravitational acceleration, and the standard deviation is slightly larger than UFK. Considering the program portability and computing power of the MCU, HBL is used for noise reduction.

Figure 3 shows the triaxial signals of the badminton drive. Each movement has its own properties, so the values of the three-axis signals are very different. Movements repeat during the acquisition time, so signals change periodically.

#### 2.3.2. Feature Extraction

Feature extraction is an important task for racquet sports recognition. To obtain optimized classification performance, the extracted features should be able to clearly represent the unique properties of movements and reduce redundancy [32]. Combining raw data, the adopted feature sets include the (1) acceleration signal magnitude vector (ASMV); (2) velocity signal magnitude vector (VSMV); (3) displacement signal magnitude vector (DSMV); (4) angle signal magnitude vector (θSMV).

ASMV is the L2 norm of the total acceleration vectors, where $a_{x}$, $a_{y}$, and $a_{z}$ denote the filtered accelerations along the x-axis, y-axis, and z-axis, respectively. This feature is independent of sensor orientation and measures the instantaneous intensity of human movements.
(3)$$ASMV=\sqrt{a_{x}{}^{2}+a_{y}{}^{2}+a_{z}{}^{2}}.$$

VSMV is the L2 norm of the velocity vectors by integrating acceleration vectors, and DSMV is obtained by integrating velocity vectors in the same way.
(4)$$VSMV=\sqrt{{\left(\int a_{x}dt\right)}^{2}+{\left(\int a_{y}dt\right)}^{2}+{\left(\int a_{z}dt\right)}^{2}}.$$

θSMV is the L2 norm of the total angle vectors. The angle obtained by the gyroscope is used as the optimum in a short time, and the average value of the angle obtained by acceleration is used to correct the angle periodically.
(5)$$\left[\begin{array}{c} \theta_{x|t}\\ \theta_{y|t}\\ \theta_{z|t} \end{array}\right]=\alpha \left[\begin{array}{c} \theta_{x|t-1}+GYRO_{x}\times T\\ \theta_{y|t-1}+GYRO_{y}\times T\\ \theta_{z|t-1}+GYRO_{z}\times T \end{array}\right]+(1-\alpha )\left[\begin{array}{c} a_{x|t}\\ a_{y|t}\\ a_{z|t} \end{array}\right],$$
(6)$$\theta SVM=\sqrt{\theta_{x}{}^{2}+\theta_{y}{}^{2}+\theta_{z}{}^{2}}.$$

### 2.4. Proposed Algorithm

The K-means algorithm uses Euclidean distance as the evaluation index of similarity and takes the compact and independent cluster as the final target. The datasets are described as $T=\left\{T_{1},T_{2},\ldots ,T_{n}\right\}$, the K cluster centers are given randomly initially, clusters can be denoted as $C_{i}=\left\{C_{1},C_{2},\ldots ,C_{k}\right\}$, and $\mu_{i}$ is the mean vector of the cluster $C_{i}$.

(7)$$\mu_{i}=\frac{1}{\left|C_{i}\right|}\sum_{x\in C_{i}}x$$

The objective function of K-means clustering is the sum of squared errors (SSE).
(8)$$SSE={\sum_{i=1}^{k}\sum_{x\in C_{i}}\left\|x-\mu_{i}\right\|}^{2}.$$

The Density-Based Spatial Clustering of Applications with Noise (DBSCAN) algorithm divides data with sufficient density into clusters, which can realize arbitrary shape clustering in noise-containing datasets. It can effectively solve the problem of misclassification caused by similar sub-features.

Both badminton and table tennis movements are combined by a variety of sub-actions. For example, a badminton drive can be decomposed into detailed actions of swinging arm and turning wrist. Therefore, manually calibrating these similar movement data is very difficult and may cause human error. The designed hybrid clustering model selects the most important macro features for different movements through a four-layer structure. The first three layers use the K-means clustering algorithm to classify and encode different movements features, decompose sub-features, and find the best clustering center for movements to distinguish them. The fourth layer uses DBSCAN to eliminate the influence of the same sub-features in different movements and determine the unique sub-features of each movement to effectively identify each movement in racquet sports. The proposed multilayer hybrid clustering model is presented in Figure 4. The output of each layer is shown in Figure 5.

The input feature vector of the first layer model is the maximum feature of the extracted feature sets after sliding sampling. The sliding window unit is the sampling point, and the sampling period of the data is 20 ms. A window with a length of 10 and a step size of 5 is used to segment the extracted feature sets to segment the movements. The dimension of the input feature vector is 120. Principal component analysis (PCA) is used to reduce the feature vector to one dimension to eliminate redundant features and reduce calculations. K-means is used to cluster feature maxima to normalize all unlabeled feature sets. The extracted features are then sorted by magnitude to label movements. 

The second layer model clusters the first layer output using K-means to obtain the sub-features decomposed in movements. The sub-features are then sorted according to frequency, which facilitates the later distinguishing of common sub-features from individual sub-features. The output of the first two-layer model is shown in Figure 5a. The x-axis indicates the features number and the y-axis indicates the movements sub-feature labels after classification and sorting.

The third layer model uses a sliding window with a length of 4 and a step size of 2 to segment the features obtained from the second layer, and takes maxima, minima, and averages of the features as inputs of the third K-means. This sliding window unit is the sub-feature point. K-means is used to divide the features in single datasets to obtain the sub-features centers of different movements. The sub-feature centers in different movement sets are shown in Figure 5b. The x-axis indicates the sub-feature labels and the y-axis indicates the sub-feature center labels. Different colors represent different movement sets (black: Walking; blue: Table tennis service; purple: Table tennis stroke; magenta: Table tennis spin; pink: Table tennis picking up; green: Badminton service; blue/black: Badminton drive; yellow: Badminton smash; orange: Badminton picking up).

The fourth layer model uses the DBSCAN algorithm to cluster the sub-feature centers obtained in the third layer. The class centers are extracted as the common features of movements, and the outliers are the exclusive features of different movements. From this, the feature center set of different movements is obtained. The target action is identified based on the normalized value of the Euclidean distance from the data output by the first three layers of clustering to the feature center, so the distance obtained by unrelated actions will be filtered when it is outside the distance threshold. Considering that the sub-feature centers obtained by clustering may overlap, a greedy algorithm is used to optimize the feature sets for obtaining the smallest feature sets of movements. The search results of the greedy algorithm are shown in Figure 5c. The x-axis indicates the number of iterations and the y-axis indicates the subsets of movement features. The sub-feature centers obtained by searching can filter wrong features, which is more accurate than the results of direct clustering. The final clustering result of the DBSCAN algorithm is shown in Figure 5d, where the *x*-axis indicates the sub-feature center labels and the I-axis indicates the predicted movement labels. The correspondence between color and movement set is consistent with 5b.

### 2.5. Software Platform

Software involves the scheduling of the model and communication in embedded platforms and a mobile phone application (App).

#### 2.5.1. Operating System

The Lite_OS operating system is ported on the hardware platform for the racquet sports recognition wristband. Lite_OS’s task module provides multi-task functions to switch and communicate between tasks. The system supports task preemptive scheduling based on priority levels and time slice rotation scheduling for the same priority. The wristband collects data in real-time and optimizes the model based on multitasking concurrent processing. The program flowchart is shown in Figure 6.

Data sampling is set to the highest priority to collect movement data in real-time. When a set of data is collected, the collection task is suspended, and the recognition task of the multilayer hybrid clustering model starts to work and analyzes the type of movement data that has just been collected. When recognition is completed, the task is suspended, and the result is then sent to the OLED screen and App through the communication task for users to view at any time. Considering that the movements of each user are slightly different, the previous movements data are stored and learned when the system is idle to continuously optimize the features in the model. The more times the user wears it, the higher the recognition accuracy of the wristband.

#### 2.5.2. Communication Protocol

Based on the TI BLE-CC254x-1.4.0 protocol stack, the management mechanism of the operating system abstraction layer (OSAL) is used to implement the Bluetooth one-master multi-slave network. The Bluetooth network automatically scans at power-on, uses MAC address matching for device screening and automatic binding, and then distinguishes the slave read–write mode according to the handle. Limited by the chip, only a maximum of 3 devices can be connected at the same time.

#### 2.5.3. App

A mobile App is designed to communicate with a wristband based on the iOS operating system. App is programmed with Xcode and accepts wristband information via Bluetooth, which is convenient for future statistics and analysis. The badminton interface is shown in Figure 7a and the table tennis interface is shown in Figure 7b. It is worth noting that information is transmitted between the wristbands via the Bluetooth network. In order to avoid breaking the current network connection when the phone is connected to wristbands as the Bluetooth master mode, App is set to connect to one wristband at the same time. 

## 3. Results and Discussion

The experimental session evaluated the model and verified the wristband in a real environment.

### 3.1. Model Evaluation

For the evaluation, three subjects’ data (60 instances) were used for training and the other two subjects’ data (40 instances) were used to test classification performance. A fivefold cross-validation guarantees that each sample point has only one chance to be classified into the training set or test set during each iteration to verify the generalization ability of the proposed model. The average accuracy of the five test results is regarded as the accuracy of the model, while the more reliable F1 score is used to evaluate the precision and recall.

There are no true labels of movements in the training dataset, so ordinary accuracy cannot be used to measure the effectiveness of the proposed model. The number of movements included in each movement instance is known, so the accuracy of the model can be evaluated by comparing the ratio of predicted movement points to the total number of points with the actual movement points to the total number of points in the dataset. Although it cannot fully characterize the accuracy of sports recognition, it can be used as a criterion for model parameter search. Combining it with the top-down greedy algorithm, it can filter the wrong features to obtain the smallest feature subset. The search results are shown in Figure 5c.

The detection results of nine movements are shown in Figure 8. The x-axis of each graph in Figure 8 indicates the features number. The upper part of each graph is the processed acceleration amplitude vector, and the lower part is the sub-feature labels obtained through the model. It can be seen from Figure 8 that each movement corresponds to a different sub-feature set after classification by the multilayer hybrid clustering model, so the model can clearly identify different movements.

Table 3 shows the classification effect of the multilayer hybrid clustering model on nine movements. The predicted proportion is the result of model classification, and the expected proportion is estimated by the number of movements and the duration of movements. Table 3 confirms the conclusion of Figure 8. The model has a higher precision for serving and picking-up movements, while the recognition precision of different hitting movements is not high, due to the small difference between them.

The normalized confusion matrix (Figure 9) shows the average evaluation results in terms of different movements.

The average accuracy in prediction is 86.32%, with an F1 score of 82.98%. Figure 9 shows that for each of these nine movements, most movements are labeled with the correct type, and the precision of some movements is more than 90%. By looking at the results in more detail, most errors can be explained. For instance, the model makes mistakes in discriminating between the different types of stroke or drive. For MCU and human observers, there are similar movements in the badminton drive and table tennis stroke. Badminton picking up and table tennis picking up also have similar movements, while badminton smash has a large amplitude in sports, which is obviously different from other movements and has a high precision. Based on the above reasons, different movements of the same kind of racquet sports are classified.

The classification results of a single racquet sport are shown in Figure 10. The accuracy of table tennis in prediction is 92.51%, with an F1 score of 92.73%. The accuracy of badminton in prediction is 94.69%, with an F1 score of 94.53%. Figure 10 shows that in the case of single racquet sport recognition, the prediction of each movement is improved. For reference only, the model in this paper was compared to similar techniques. Martin et al. [6] proposed a Siamese spatiotemporal convolution (SSTC) method based on the RGB image sequence and its calculated optical flow to classify table tennis strokes. The accuracy of this method was 91.4%. The model proposed in this paper has a competitive accuracy in table tennis movement recognition. Meanwhile, compared to video recognition based on large data streams, the method based on motion sensors has more advantages in computing time and storage costs. Wang et al. [28] used a SVM to recognize the information collected by the motion sensor and were able to recognize three different badminton strokes. The accuracy of the system based on SVM was 94%, and the accuracy of the system based on PCA + SVM was 97%. The accuracy of the model proposed in this paper is slightly lower than that of the above model in badminton movement recognition. The advantage of this proposed system is that movement can be recognized on the wristband in real-time.

#### 3.1.1. Comparison of Different Cluster Numbers

The model ranks the features of the first two layers of output, so features are highly consistent after multi-layer clustering, and the model accuracy is not affected by the randomness of the K-means algorithm. The first layer determines the number of sub-features, so the accuracy is mainly affected by the number of clusters in the first layer. Table 4 shows the different K values and their corresponding accuracy. The best classification effect is obtained when the number of clusters is 70. A smaller number of clusters will cause feature overlap, which will increase the difficulty of subsequent sub-feature discrimination. A larger number of clusters will increase noise, so the cluster numbers need to be adjusted to appropriate parameters.

#### 3.1.2. Comparison of Different Classifiers

We compared the linear discriminant function (LDF), random forest, SVM, and multilayer hybrid clustering model proposed in this paper, as shown in Table 5. The time in Table 5 is the training time of each classifier, and it can clearly reflect the computational cost. The model proposed in this paper achieves the best performance in terms of recognition accuracy and training time.

### 3.2. Wristband Verification

A tester was randomly selected to wear the wristband for badminton and table tennis sports tests. The recognition time of the wristband includes the data collection time, algorithm recognition time, and transmission time to the OLED screen. The recognition time for different movements is different. The average recognition time of the wristband is about 1 s. The Bluetooth transmission rate is 115,200 bps, and the communication time for the wristband to the mobile phone is about 0.5 s. The time from wristband recognition to displaying the result on the mobile phone is about 1.5 s. Basically, the recognition result can be obtained on the wristband after completing one movement. The proposed wristband realizes real-time recognition of the racquet sports. The recognition results of the wristband on badminton and table tennis are shown in Figure 11a,b respectively. The test accuracy is shown in Table 6. Experiments have verified the feasibility of the racquet sports recognition wristband, and the average accuracy is above 77%.

## 4. Conclusions

This paper presents a racquet sports recognition system to effectively sense movement parameters. The system consists of an IMU, BLE technology, mobile application, and multi-layer hybrid clustering model. The wristband uses the integrated IMU to obtain movement data, and then runs the recognition model and performs the number counting in the MCU. The communication between phone and wristband and the networking of multiple devices are realized through the Bluetooth mesh network, which is convenient for users to track their exercise through the App and provides information-sharing functions for players and referees to improve the fairness of the game.

A multilayer hybrid clustering model similar to neural networks (NN) is proposed to improve recognition accuracy. The multi-layer K-means clustering algorithm is used for feature extraction and segmentation, and the DBSCAN is used to further classify features with the same sub-features. The model can identify unlabeled and noisy data without data calibration, which enables the sensor system to achieve greater calculation and lower energy.

The experimental results confirm that the racquet sports recognition wristband designed in this paper can obtain various effective movements features from the wrist and classify badminton and table tennis. The accuracy of the model decreases slightly in practice, but the wristband recognition results are basically consistent with actual movements. Compared to the machine vision-based methods, wristbands have great advantages in terms of privacy protection and tolerance to external environments. The wristband provides a reference solution for the commercial application of racquet sports recognition. 

The dataset in this paper is mainly composed of target movements, so limiting the use of the wristband to racquet sports will have higher accuracy. In wider usage scenarios, such as daily activities, misidentification may be caused because some non-target movements have the same sub-features as target movements. In the next stage, more types of movement data will be collected to improve the accuracy of the model and evaluate more movement details.

## Figures and Tables

**Figure 1 sensors-20-01638-f001:**
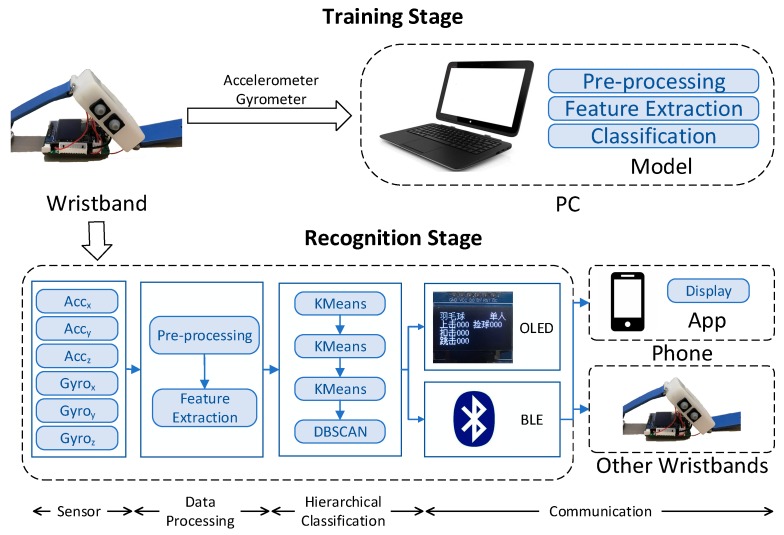
Proposed system architecture. In the training stage, the wristband collects datasets and the model feasibility is verified on a personal computer (PC). In the recognition stage, data are acquired and identified in the wristband, and the system communicates via Bluetooth.

**Figure 2 sensors-20-01638-f002:**
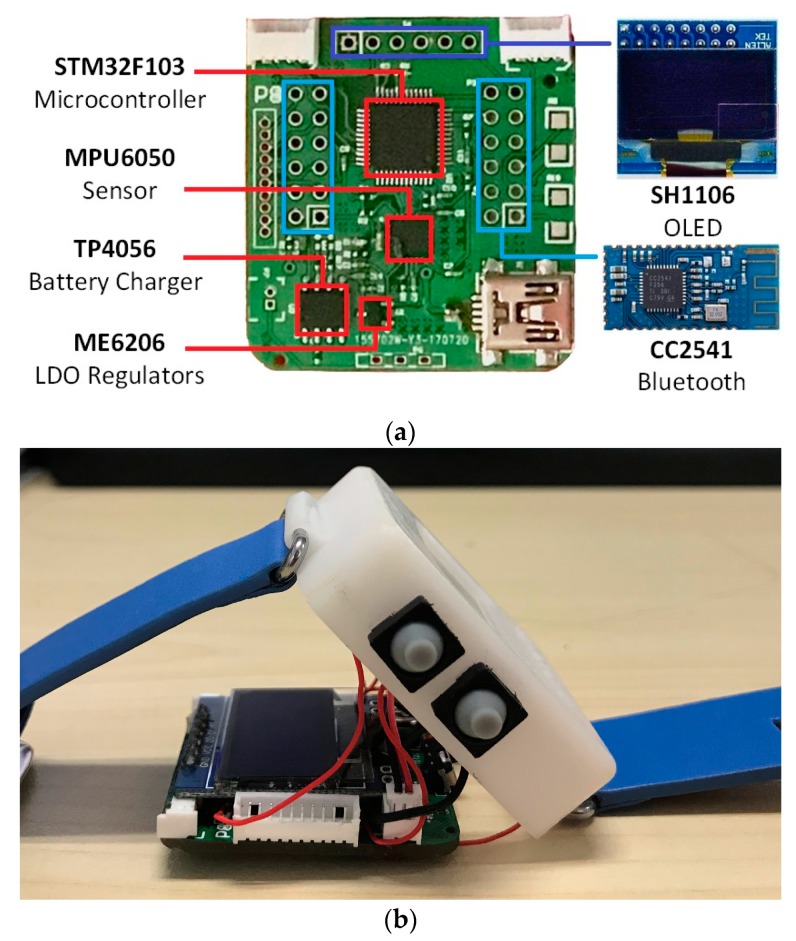
(**a**) Circuit board of integrated wearable sensor platform. The organic light-emitting diode (OLED) screen and Bluetooth are modular structures, plugged into the board. (**b**) Overall structure of the wristband.

**Figure 3 sensors-20-01638-f003:**
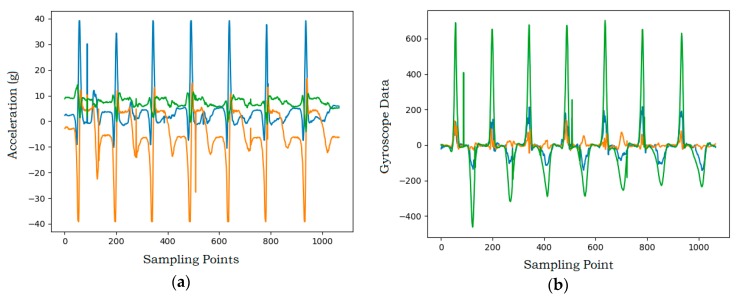
Triaxial signals of badminton drive. (**a**) Triaxial acceleration signals for badminton drive; (**b**) triaxial angular velocity signals for badminton drive.

**Figure 4 sensors-20-01638-f004:**
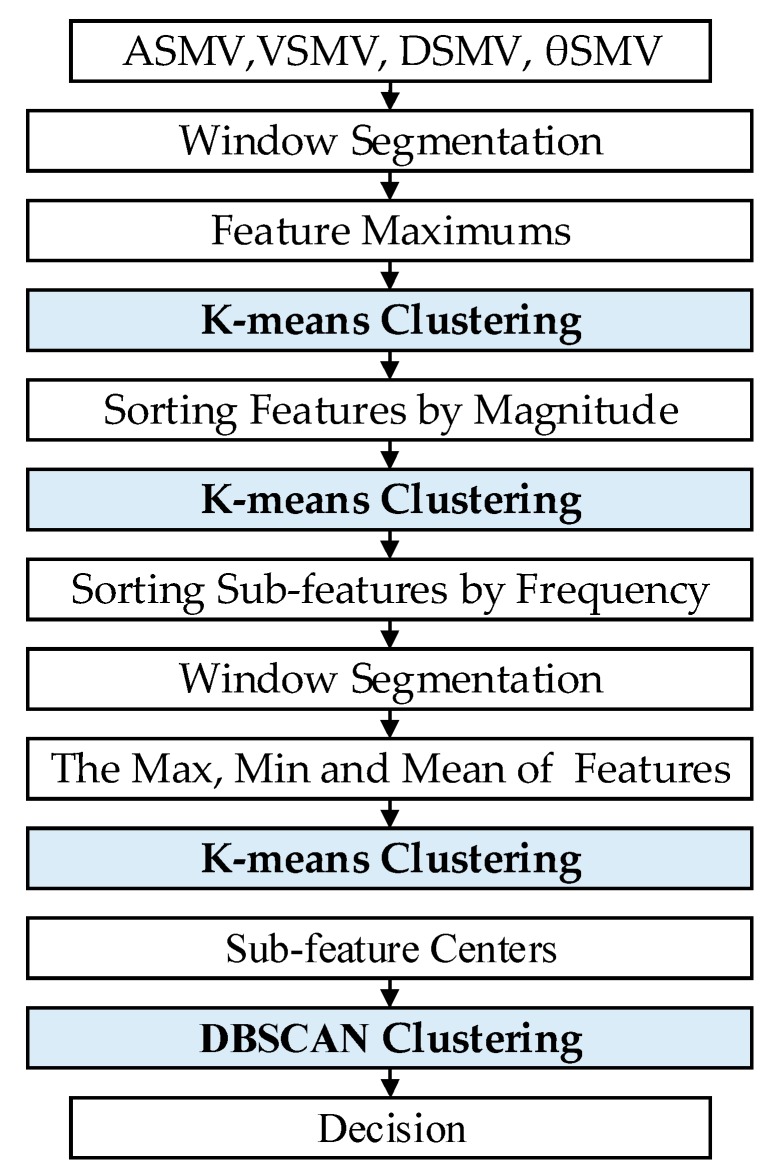
The proposed multilayer hybrid clustering model framework.

**Figure 5 sensors-20-01638-f005:**
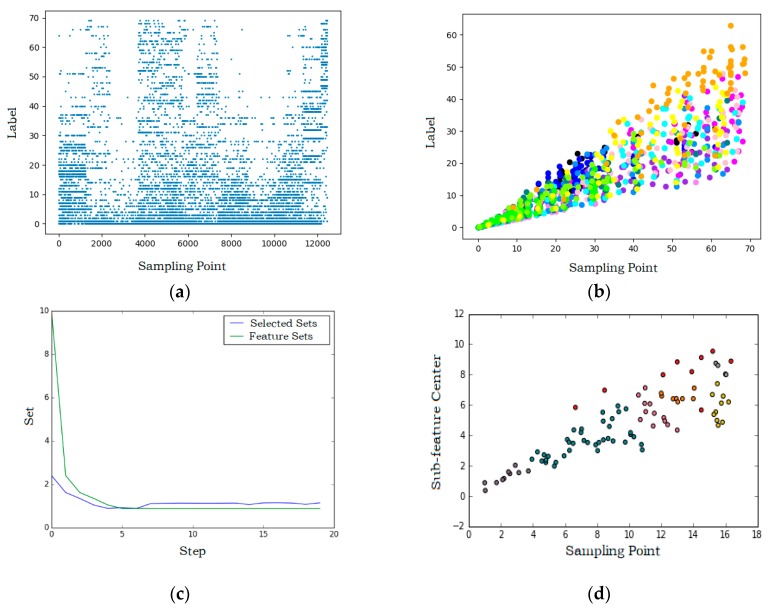
Each layer output of the model. (**a**) All movements sub-features after classification and sorting; (**b**) the sub-feature centers of nine movements are extracted separately and expressed in different colors; (**c**) minimum sub-feature set obtained by greedy algorithm; (**d**) sub-feature centers obtained through the density-based spatial clustering of applications with noise (DBSCAN).

**Figure 6 sensors-20-01638-f006:**
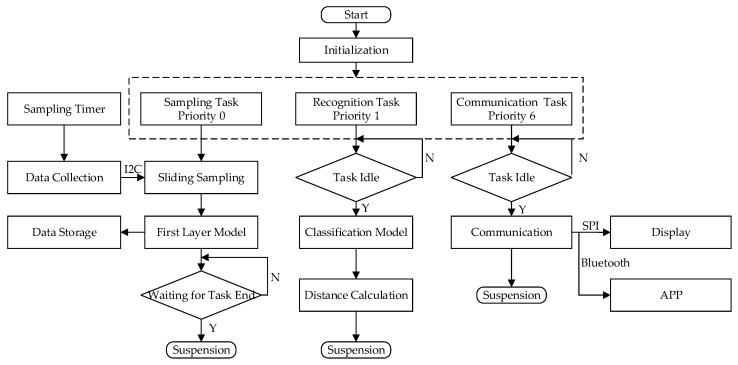
The program flowchart.

**Figure 7 sensors-20-01638-f007:**
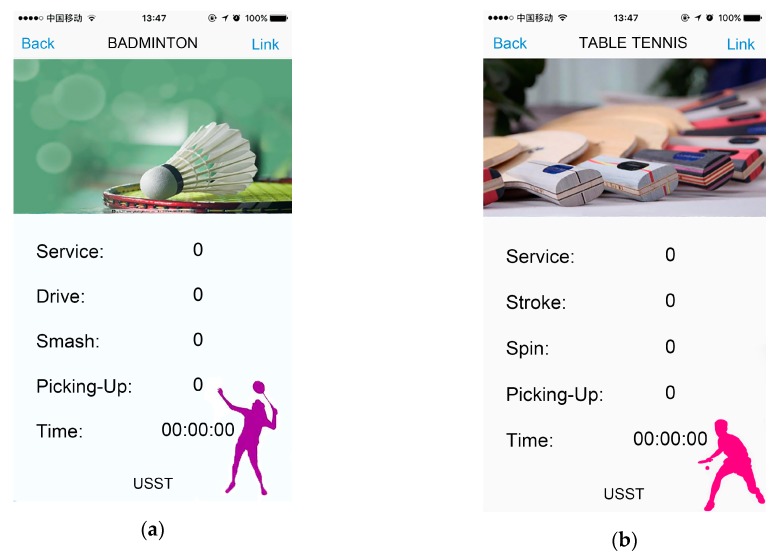
(**a**) The badminton interface on App; (**b**) the table tennis interface on App. Both interfaces contain 5 types of information, including service times, drive times, smash times, picking up times, and time.

**Figure 8 sensors-20-01638-f008:**
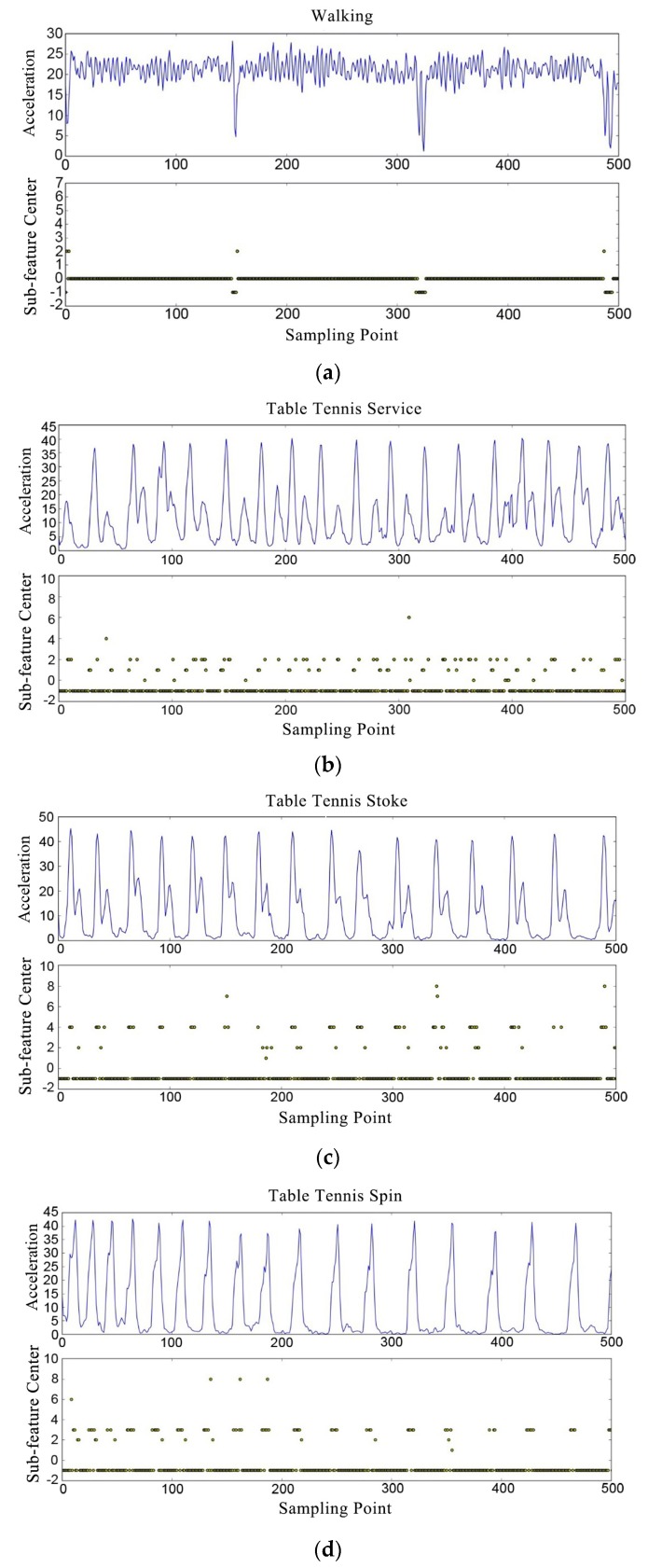
The movements and corresponding detection effect. (**a**) walking; (**b**) table tennis service; (**c**) table tennis stoke; (**d**) table tennis spin; (**e**) table tennis picking up; (**f**) badminton service; (**g**) badminton drive; (**h**) badminton smash; (**i**) badminton picking up.

**Figure 9 sensors-20-01638-f009:**
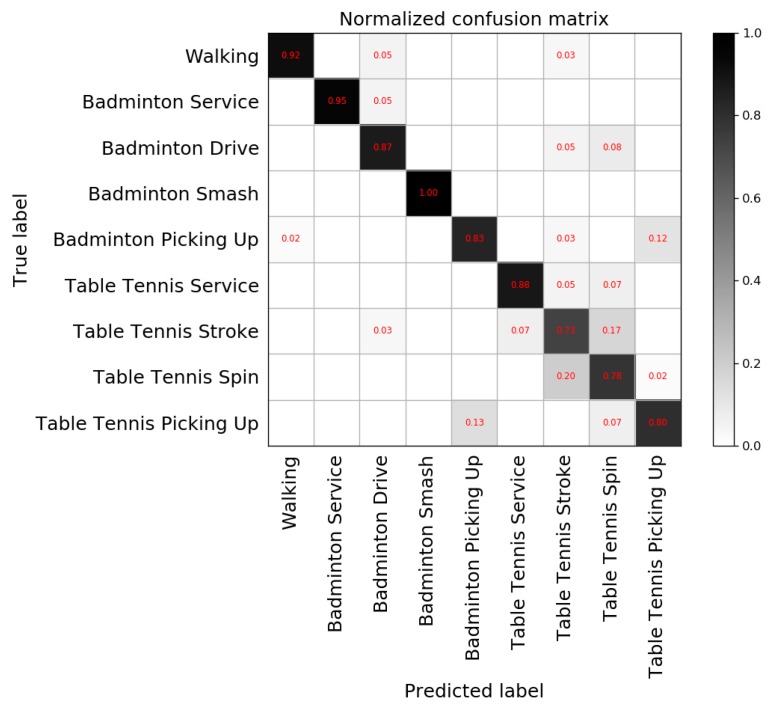
The normalized confusion matrix of the proposed model.

**Figure 10 sensors-20-01638-f010:**
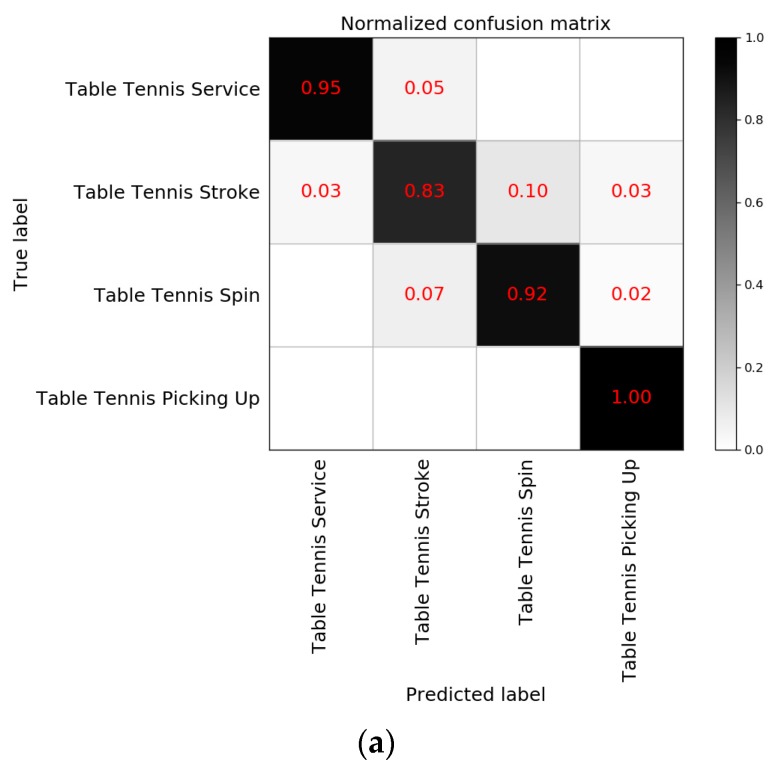
(**a**) The normalized confusion matrix of table tennis; (**b**) the normalized confusion matrix of badminton.

**Figure 11 sensors-20-01638-f011:**
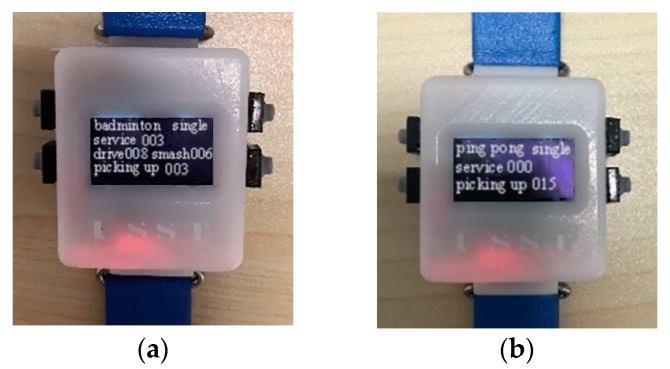
(**a**) Badminton movements recognition results; (**b**) table tennis movements recognition results.

**Table 1 sensors-20-01638-t001:** The parameters of the wristband.

Size	Voltage Level	Screen Resolution	Gyro Sensitivity	Accel Sensitivity
38 mm × 34 mm × 20 mm	3.3 V	128 × 64 pixels	16.4 LSB/°/s	8192 LSB/g

**Table 2 sensors-20-01638-t002:** Mean and standard deviation of acceleration in different filters.

Filter	Mean	Standard Deviation
**HBL**	9.489730	0.013882
**UKF**	8.921552	0.010730

**Table 3 sensors-20-01638-t003:** The classification effect of the multilayer hybrid clustering model on 9 movements.

Category	Expected Proportion	Forecast Proportion
Walking	0.76471	0.71649
Table Tennis Service	0.19161	0.19161
Table Tennis Stroke	0.23232	0.17676
Table Tennis Spin	0.17241	0.22413
Table Tennis Picking Up	0.20623	0.24098
Badminton Service	0.21038	0.21818
Badminton Drive	0.21379	0.15862
Badminton Smash	0.20712	0.20388
Badminton Picking Up	0.23529	0.20941

**Table 4 sensors-20-01638-t004:** Precision of different cluster numbers.

Cluster Numbers	Precision
20	0.45
50	0.77
70	0.86
100	0.62

**Table 5 sensors-20-01638-t005:** Average recognition accuracy and time of different classification algorithms.

Classification Algorithm	Accuracy	Time(s)
Linear Discriminant Function (LDF)	0.69	6.38
Random Forests	0.74	18.25
SVM	0.76	145.54
Proposed Model	0.86	27.73

**Table 6 sensors-20-01638-t006:** The wristband recognition accuracy.

Category	Accuracy
Walking	0.91
Table Tennis Service	0.79
Table Tennis Stroke	0.68
Table Tennis Spin	0.71
Table Tennis Picking Up	0.77
Badminton Service	0.82
Badminton Drive	0.63
Badminton Smash	0.89
Badminton Picking Up	0.75

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
