# Peer review of "Racquet Sports Recognition Using a Hybrid Clustering Model Learned from Integrated Wearable Sensor"

_sensors, 2020, doi:10.3390/s20061638_

Round 1

Reviewer 1 Report

The paper describes a model and implementation for recognition of different movements during racquet sports (tennis and badminton) based on Inertial sensor signals. The model is based on rotation-invariant features and clustering. The results show high accuracy in the recognition when testing. The paper has been improved greatly and it has addressed most of my concerns. However, more details are needed in the evaluation description and method section. 

First, it is not clear how the different sliding windows are composed and what is the length of the window used the final analysis and how often is a recognition decision made.

For instance, in Line 156, what is the unit of the window size? Seconds? Samples?

Then, in line 197, there is another sliding window: is this the same? Or is this a window over the previous window? What is the unit of this window then? Same for the third layer window (Line 208)

It is important to understand this as it has a clear impact on the real-time recognition objective.

As for the evaluation, in line 287 it is said that 5 subjects performed 20 trials for 9 movements. Is it 20 trials for each movement? Or in 20 trials they performed all movements? More details are needed on the data. How was it labeled? How long did it take to complete a movement? Was the movement continuously done? Was there a sequence of movements? Did all participants follow the same sequence? Are the users experts or novices in each sport? Can all users play both sports? How many windows were finally obtained from the data?

This affects the evaluation, and the evaluation results severely and thus can not be omitted. 

In line 290 you say: “Fivefold 290 cross-validation is used to avoid overfitting problems.” It is not clear when the cross-validation was used it he dataset was split into training and testing. Was it for parameter optimization? If so, which parameters were optimized?

Analysis of your objective, is this system suitable for instantaneous analysis and data sharing on wearable devices? In other words, is the average time of 27.73 seconds (if I understood correctly) enough for real-tie recognition compared to the duration of the movements?

Line 281 - 283 “The ratio of each movement to the total number was used during the training phase to determine the precision of the model. Although it cannot fully characterize the accuracy of sports recognition, it can be used as a criterion for evaluating model results.” Please explain why this can be used as a measure for evaluation. What are the requirements of the system that make it an appropriate measure?

In the conclusions, it would be good to highlight that not only the model but also the whole system was designed, including the sensor, communications and application.

Please check the format of the references. References 6, 7, 8, 9, 11, 14, 19, 20, 24, 29 are written with title in italics

Author Response

Point 1: The paper describes a model and implementation for recognition of different movements during racquet sports (tennis and badminton) based on Inertial sensor signals. The model is based on rotation-invariant features and clustering. The results show high accuracy in the recognition when testing. The paper has been improved greatly and it has addressed most of my concerns. However, more details are needed in the evaluation description and method section.

 Response 1: We thank the reviewer for the positive comments and supplemented the method and evaluation sections in the article based on the reviewer's comments. For more detailed point-by-point responses, please see below.

Point 2: First, it is not clear how the different sliding windows are composed and what is the length of the window used the final analysis and how often is a recognition decision made.

For instance, in Line 156, what is the unit of the window size? Seconds? Samples?

Then, in line 197, there is another sliding window: is this the same? Or is this a window over the previous window? What is the unit of this window then? Same for the third layer window (Line 208)

It is important to understand this as it has a clear impact on the real-time recognition objective.

Response 2: The above questions are all about the window of the model, so we summarize them together and respond.

This article uses two sliding windows. Our idea is to use the first sliding window to divide the movement into sub-grained sub-movements. A movement is composed of multiple sub-movements, so a second sliding window is used to aggregate the sub-movements into more complex process features. The model proposed in this paper takes about 1s to recognize one movement.

The time for the subject to complete a movement is about 1.2s. The sampling frequency of the model is 50Hz. In the first sliding window, we use a window with a length of 10 and a step size of 5 to segment the sampling points, and the sliding window unit is the sampling point. In the original text, the sliding window in line 156 is the same as the sliding window in line 197, and both are the descriptions of the first sliding window. To avoid misunderstanding to the reader, only the introduction of the first sliding window in line 197 (now line 201) is retained in the revised version.

The second sliding window uses a window with a length of 4 and a step size of 2 to segment the sub-feature after clustering analysis to obtain the characteristics of a movement. The second sliding window unit is the sub-feature point, so the window length used in the final analysis is 4 sub-feature points.

In the revised manuscript, we added a description of the window based on the reviewer's suggestions, as follows:

“The input feature vector of the first layer model is the maximum feature of the extracted feature sets after sliding sampling. The sliding window unit is the sampling point, and the sampling period of the data is 20ms. A window with a length of 10 and a step size of 5 is used to segment the extracted feature sets to segment the movements. The dimension of the input feature vector is 120.”

“The third layer model uses a sliding window with a length of 4 and a step size of 2 to segment the features obtained from the second layer, and takes maximums, minimums and averages of features as input of the third K-means. This sliding window unit is the sub-feature point.”

Point 3: As for the evaluation, in line 287 it is said that 5 subjects performed 20 trials for 9 movements. Is it 20 trials for each movement? Or in 20 trials they performed all movements? More details are needed on the data. How was it labeled? How long did it take to complete a movement? Was the movement continuously done? Was there a sequence of movements? Did all participants follow the same sequence? Are the users experts or novices in each sport? Can all users play both sports? How many windows were finally obtained from the data?

This affects the evaluation, and the evaluation results severely and thus can not be omitted.

 Response 3: The above questions are all about dataset, so we summarize them together and respond.

A total of 5 subjects participated in the collection of 9 target movements. Each subject performed 20 tests for each movement.

During the experiment, we manually recorded the number of movements in each movement set to label the dataset.

For a subject, the time to complete different movements is different, and the time to complete one movement is within 1s to 1.2s.

We continuously collect the movements in the same movement set and collect different movement sets separately, which is convenient for labelling the actions, so we have not collected the combined data of the target action. Since the movements of different movement sets are collected separately and the data is recorded in real time during the movement, we consider that the sequence of movement does not affect the datasets.

Among the five subjects, one subject has received 2 years of professional training in badminton, and one subject has received 4 years of professional training in table tennis. Others are amateurs.

100 instances were collected for each movement set, and a total of 900 instances were collected.

The introduction of the data set in the original text is scattered in the data collection section and evaluation section, so it is not easy for the reader to understand. In the revised manuscript, we integrate the introduction of the datasets into the 2.2 Data Collection section and added a specific introduction to the dataset based on the reviewer's suggestions, as follows:

“A total of 5 healthy subjects (3 males, 2 females; age: 25 ± 5) took part in the data collection process. Among them, one subject has received 2 years of professional training in badminton, and one subject has received 4 years of professional training in table tennis. Others are untrained people. All participants provided written informed consent before participation. Subjects were asked to wear the racquet sports recognition wristband in their dominant wrist. Subjects are all right-handed. The datasets were collected in real training environment (gym).

In experiment, each subject performed nine kinds of movements: four types of table tennis (Service, Stroke, Spin, and picking up), four types of badminton (Service, Drive, Smash, and picking up) and walking. Each subject performed 20 tests for each movement. For a subject, the time to complete different movements is different, and the time to complete one movement is within 1s to 1.2s. The action in the same movement set was collected continuously, and different movement sets were collected separately. 100 instances were collected for each movement set, and a total of 900 instances were collected. During the experiment, the number of actions in each movements set was manually recorded to label the data set at a later stage.”

Point 4: In line 290 you say: “Fivefold 290 cross-validation is used to avoid overfitting problems.” It is not clear when the cross-validation was used it he dataset was split into training and testing. Was it for parameter optimization? If so, which parameters were optimized?

Response 4: Fivefold cross-validation is a common method for verifying the generalization ability of a model. It is used to divide the datasets at the beginning. It can completely separate the training set and the test set to prevent the data of the training set and the test set from intersecting and causing falsely high accuracy. Therefore, it is not used for parameter optimization in this paper, but used to ensure the reliability of the accuracy of the model.

Considering that the introduction of cross-validation in the middle of the evaluation section of the original text will mislead the reader. In the modified version we put it at the beginning of the evaluation section and added an introduction to cross-validation as follows:

“For the evaluation, three subjects’ data (60 instances) were used for training and the other two subjects’ data (40 instances) were used to test classification performance. Fivefold cross validation guarantees that each sample point has only one chance to be classified into the training set or test set during each iteration to verify the generalization ability of the proposed model. The average accuracy of the five test results is regarded as the accuracy of the model, while the more reliable F1 score is used to evaluate the precision and recall.”

 Point 5: Analysis of your objective, is this system suitable for instantaneous analysis and data sharing on wearable devices? In other words, is the average time of 27.73 seconds (if I understood correctly) enough for real-tie recognition compared to the duration of the movements?

 Response 5: In fact, the time (27.73 seconds) mentioned in Table 4 is the training time of the model. The purpose of this table is to show that the model proposed in this paper has an advantage over other models in training time. The recognition time of the wristband includes the data collection time, the algorithm recognition time and the transmission time to the LED screen. The recognition time for different movements is different. The average recognition time of the wristband is about 1s. Basically, the recognition result can be obtained on the wristband after completing one movement.

In the revised manuscript, we added an introduction to recognition time in the 3.2 Wristband Verification section, as follows:

“The recognition time of the wristband includes the data collection time, the algorithm recognition time and the transmission time to the LED screen. The recognition time for different movements is different. The average recognition time of the wristband is about 1s. The Bluetooth transmission rate is 115200bps, and the communication time for the wristband to the mobile phone is about 0.5s. The time from the wristband recognition to displaying the result on the mobile phone is about 1.5s. Basically, the recognition result can be obtained on the wristband after completing one movement. The proposed wristband realizes the real-time recognition of the racquet sports.”

 Point 6: Line 281 - 283 “The ratio of each movement to the total number was used during the training phase to determine the precision of the model. Although it cannot fully characterize the accuracy of sports recognition, it can be used as a criterion for evaluating model results.” Please explain why this can be used as a measure for evaluation. What are the requirements of the system that make it an appropriate measure?

 Response 6: Our model uses multilayer hybrid clustering to recognize movements. During the clustering process, the true labels of the data do not participate in training. We need a simple and effective discrimination method to verify the effect of clustering. The number of movements contained in each movement instance is known, so we simply evaluate the model by comparing the ratio of predicted movement points to the total number and the ratio of actual movement points to the total number in the datasets. Meanwhile, we use the result of this method as a criterion for optimizing parameters. We combine it with the top-down greedy algorithm to filter duplicate features and search to obtain the smallest feature subset. The results obtained on the test set have the same trend as the results obtained by this method, which validates this evaluation method.

Based on reviewer's comments, we have expanded the introduction of the simple evaluation model in the evaluation section of the revised manuscript, as follows:

“There are no true labels of movements in the training dataset, so ordinary accuracy cannot be used to measure the effectiveness of the proposed model. The number of movements included in each movement instance is known, so the accuracy of the model can be evaluated by comparing the ratio of predicted movement points to the total number of points with the actual movement points to the total number of points in the dataset. Although it cannot fully characterize the accuracy of sports recognition, it can be used as a criterion for model parameter search. Combining it with the top-down greedy algorithm can filter the wrong features to get the smallest feature subset. The search results are shown in Figure 5 (c).”

 Point 7: In the conclusions, it would be good to highlight that not only the model but also the whole system was designed, including the sensor, communications and application.

 Response 7: We thank the reviewers for their suggestions on the conclusion section of this article. In the revised manuscript, we added an introduction to the wristband system in the conclusion section as follows:

“This paper presents a racquet sports recognition system to effectively sense movement parameters. The system consists of IMU, BLE technology, mobile application and multi-layer hybrid clustering model. The wristband uses the integrated IMU to obtain movement data, then runs the recognition model and do the number counting in the MCU. The communication between phone and wristband and the networking of multiple devices are realized through the Bluetooth mesh networking, which is convenient for users to track their exercise through the App and provides information-sharing functions for players and referees to improve the fairness of the game.”

Point 8: Please check the format of the references. References 6, 7, 8, 9, 11, 14, 19, 20, 24, 29 are written with title in italics.

Response 8: We apologize for the formatting errors in the references in the original manuscript and thank you for your correction. In the revised manuscript, we obtained the MDPI Chicago Style EndNote template file from the website and modified the format of the references based on this template. The format of the reference is modified as follows:

Reviewer 2 Report

The reviewer appreciates the authors for revising the manuscript and improving its quality. The authors have answered most of the concerns that the reviewer raised over the previous version of the paper. There are, however, few important aspects of the proposed method and system that must be cleared or incorporated to make the paper sound, technically, experimentally, and presentation wise.

There are numerous existing similar techniques that can perform the job well, some are able to to achieve the similar or even better results (reported in the respective papers) than the proposed system. Moreover, some of them are purely video-based method that do not require the specific expensive sensors to work. The reviewer mentioned a number of papers in his/her last review of the manuscript. To justify the contribution(s) of present results, the again wishes to see the results of the proposed method compared with few similar techniques. This might put the authors to capture some more dataset, but it would certainly increase the credibility of the method and its effectiveness.

The experimental evaluation is performed with F score. The reviewer would suggest the authors to use more parametric and non-parametric measures to evaluate the performance of the proposed method.

An analysis of the computational complexity of the proposed system could be interesting for the readers. A time complexity analysis of the system with different classifier is though presented in Section 3.1.2, but it reports the training time. What is testing time? That is, does the proposed system able to work real time? If no, then how much delay occurs in testing a signal?

Author Response

Point 1: The reviewer appreciates the authors for revising the manuscript and improving its quality. The authors have answered most of the concerns that the reviewer raised over the previous version of the paper. There are, however, few important aspects of the proposed method and system that must be cleared or incorporated to make the paper sound, technically, experimentally, and presentation wise.

Response 1: We thank the reviewer for the positive comments and have revised the evaluation section of this paper based on the comments. For more detailed point-by-point responses, please see below.

Point 2: There are numerous existing similar techniques that can perform the job well, some are able to achieve the similar or even better results (reported in the respective papers) than the proposed system. Moreover, some of them are purely video-based method that do not require the specific expensive sensors to work. The reviewer mentioned a number of papers in his/her last review of the manuscript. To justify the contribution(s) of present results, the again wishes to see the results of the proposed method compared with few similar techniques. This might put the authors to capture some more dataset, but it would certainly increase the credibility of the method and its effectiveness.

Response 2: In the introduction section, we introduced many similar articles on identifying racquet movements, including those we collected and introduced by previous reviewers. We add articles mentioned by reviewers to references to increase the persuasiveness of this paper. Different technologies focus on different directions. Video-based methods can achieve more fine-grained motion recognition, but image-based datasets have large data flows and large amounts of calculations, which is not conducive to real-time recognition of movement. Based on the above reasons, we obtain movement characteristics through IMUs. The above articles mainly recognize different movements of a single racquet sport in the experiment, and lacks the recognition results of multiple racquet sports. Therefore, providing a system that can recognize multiple racquet sports in real-time is the main contribution of this paper. This paper extended existing sensor-based movement measuring methods with a multilayer hybrid clustering model.

We carefully considered the reviewer's suggestions. The video-based method and the IMUs-based method use different datasets. Our method cannot realize the identification of the video dataset, and the video-based method cannot be used in our dataset for validation. Based on the above reasons, we compared the accuracy of this article with the accuracy provided in a video-based article identifying table tennis and an IMUs-based article identifying badminton. We have modified the evaluation section of the manuscript, as follows:

“For reference only, the model in this paper is analysed with similar techniques. Martin et al. [6] proposed a Siamese spatio-temporal convolution (SSTC) method based on the RGB image sequence and its calculated optical flow to classify table tennis strokes. The accuracy of this method is 91.4%. The model proposed in this paper has a competitive accuracy in table tennis movement recognition. Meanwhile, compared with video recognition based on large data streams, the method based on motion sensors has more advantages in computing time and storage costs. Wang et al. [28] used SVM to recognize the information collected by the motion sensor and were able to recognize three different badminton strokes. The accuracy of the system based on SVM is 94%, and the accuracy of the system based on PCA + SVM is 97%. The accuracy of the model proposed in this paper is slightly lower than the above model in of badminton movement recognition. The advantage of this proposed system is that movement can be recognized on the wristband in real-time.”

Point 3: The experimental evaluation is performed with F score. The reviewer would suggest the authors to use more parametric and non-parametric measures to evaluate the performance of the proposed method.

Response 3: We thank the reviewer for the suggestions. We use a number of methods to evaluate the model directly or indirectly.

Fivefold cross-validation is a common method for verifying the generalization ability of a model. It is used to divide the datasets at the beginning. It can completely separate the training set and the test set to prevent the data of the training set and the test set from intersecting and causing falsely high accuracy.

During training, we simply evaluate the model by comparing the ratio of predicted movement points to the total number and the ratio of actual movement points to the total number in the datasets. Meanwhile, we use the result of this method as a criterion for optimizing parameters. We combine it with the top-down greedy algorithm to filter duplicate features and search to obtain the smallest feature subset. The results obtained on the test set have the same trend as the results obtained by this method, which validates this evaluation method.

In the evaluation of the test set, we used the confusion matrix, accuracy, and F1-score to verify the model. Confusion matrix is a standard format for accuracy evaluation. Each column of the confusion matrix represents the prediction category, and each row represents the true category of the data. Accuracy is the most common metric for model validation. F1-score takes into account both the accuracy and recall of the classification model, and can evaluate the model more comprehensively.

Finally, we considered the reviewer’s suggestions and compared the accuracy of this article with the accuracy provided in a video-based article identifying table tennis and an IMUs-based article identifying badminton.

Point 4: An analysis of the computational complexity of the proposed system could be interesting for the readers. A time complexity analysis of the system with different classifier is though presented in Section 3.1.2, but it reports the training time. What is testing time? That is, does the proposed system able to work real time? If no, then how much delay occurs in testing a signal?

Response 4: The recognition time of the wristband includes the data collection time, the algorithm recognition time and the transmission time to the LED screen. The recognition time for different movements is different. The average recognition time of the wristband is about 1s. The time for the subject to complete a movement is about 1.2s. Basically, the recognition result can be obtained on the wristband after completing one movement.

In the revised manuscript, we added an introduction to recognition time in the 3.2 Wristband Verification section, as follows:

“The recognition time of the wristband includes the data collection time, the algorithm recognition time and the transmission time to the LED screen. The recognition time for different movements is different. The average recognition time of the wristband is about 1s. The Bluetooth transmission rate is 115200bps, and the communication time for the wristband to the mobile phone is about 0.5s. The time from the wristband recognition to displaying the result on the mobile phone is about 1.5s. Basically, the recognition result can be obtained on the wristband after completing one movement. The proposed wristband realizes the real-time recognition of the racquet sports.”

Reviewer 3 Report

I am the reviewer 1 in the previous submission. I acknowledge significant effort to improve the paper. I think the revised paper addressed my previous concerns. There are one major issue and a few minor issues:

In the real environment, there are numerous activities that are not related to the target activities, e.g., standing still, picking up towel, drinking from a bottle of water, etc. To make the wearable sports activity recognition system useful, such unrelated activities must be eliminated, which seems missing in the current system. The authors needs to mention this and present possible solution for this issue.

In line 253: "Mac" address should be "MAC" address. 

In line 257: "IOS" should be "iOS".

Author Response

Point 1: I am the reviewer 1 in the previous submission. I acknowledge significant effort to improve the paper. I think the revised paper addressed my previous concerns. There are one major issue and a few minor issues:

Response 1: We thank the reviewer for the positive comments and supplemented the method and conclusions sections in the article based on the reviewer's questions. For more detailed point-by-point responses, please see below.

Point 2: In the real environment, there are numerous activities that are not related to the target activities, e.g., standing still, picking up towel, drinking from a bottle of water, etc. To make the wearable sports activity recognition system useful, such unrelated activities must be eliminated, which seems missing in the current system. The authors needs to mention this and present possible solution for this issue.

Response 2: The fourth layer mode determines the target movement based on the normalized value of the Euclidean distance from the data output by the first three layers of clustering to the feature center. Therefore, when the distance obtained by some unrelated movements exceeds the distance threshold, the model will filter out these unrelated actions. The wristband can filter out most irrelevant movements in the scene of racquet sports. However, if its application scenario is set to a wider range (such as daily activities), some non-target movements may cause misidentification because they may have the same sub-functions as the target movement.

In the revised manuscript, we added our model's handling of irrelevant actions in the 2.4 Proposed Algorithm section based on reviewers' comments and mentioned the limitations of the model in the conclusion section, as follow:

“The fourth layer model uses the DBSCAN algorithm to cluster the sub-feature centers obtained in the third layer. The class centers are extracted as the common features of movements, and the outliers are the exclusive features of different movements. From this, the feature center set of different movements are obtained. The target action is identified based on the normalized value of the Euclidean distance from the data output by the first three layers of clustering to the feature center, so the distance obtained by unrelated actions will be filtered when it is outside the distance threshold.”

“The dataset in this paper is mainly composed of target movements, so limiting the use of the wristband to racquet sports will have higher accuracy. In wider usage scenarios, such as daily activities, misidentification may be caused because some non-target movements have the same sub-features as target movements. In the next stage, more types of movement data will be collected to improve the accuracy of the model and evaluate more movement details.”

Point 3: In line 253: "Mac" address should be "MAC" address.

In line 257: "IOS" should be "iOS".

Response 3: Thanks to the reviewer for correcting the spelling errors in the original manuscript, we have corrected them in lines 261 and 265 of the revised manuscript, as follow:

“Bluetooth network automatically scans at power-on, uses MAC address matching for device screening and automatic binding, and then distinguishes slave read-write mode according to handle.”

“A mobile App designed to communicate with a wristband based on the iOS operating system.”

Round 2

Reviewer 1 Report

The authors have replied to all my comments and concerns adequately and the paper has improved.  

Reviewer 2 Report

The authors have answered/incorporated most of the reviewer's comments. The manuscript, however, requires language proofreading.

The reviewer has no more comments.

This manuscript is a resubmission of an earlier submission. The following is a list of the peer review reports and author responses from that submission.

Round 1

Reviewer 1 Report

The paper shows a recognition system for racquet sports actions, which consists of a sensing HW&SW platform and a recognition method based on cascaded clustering. 

First of all, it is not clear the contributions with challenges. In lines 74-80, one of contributions is described, which is about the wristband type recognition system. Actually, in Section 2.1 and 2.5, hardware and software platforms are designed; however, why does it need to be designed, rather than using a commercial smartwatch? What is special requirement in racquet sports activity recognition? If no specific requirement exist, the first contribution is not actually a contribution. Although a sentence in lines 70-72 says that existing technology only recognizes racquet sports movement through analysis on PC platform, it does not show that running analysis on smartwatch platform is impossible, in which the evolution of the recognition method might be the focus of the work. So, the paper should clearly and reasonably state the contribution. 

Regarding the experiments, it is not clear how the classification test was carried out. For example, how the training and test data are split: N-fold cross-validation or leave-one-person-out cross validation? How many instances of feature vector (windows) are used? Also, the classified result should be analyzed in more detail. For example, the tendency of misclassification can be understood using a confusion matrix. Additionally, how robust the method is among individuals? Furthermore, in Section 4, the manuscript say that the number of the clusters in each layer is important factor for the accuracy; however, there is no comparison using various K-values, which must be shown in the experimental section and then discussed. 

Another important missing aspect is the baseline to show the proposed approach, i.e., hybrid clustering-based method, is suitable for the racquet activity recognition because lines 61-65 state that traditional machine learning approach has difficulty in recognizing similar movements often found in racquet sports. To validate the proposed method, test with traditional ML method such as SVM, Decision Tree, and RandomForests should be carried out, which I consider an offline-based test on a PC using an existing library such as scikit-learn for Python is enough.

Regarding the presentation, the figures (graphs) should be clearly explained. For example, what the axes in Figures 6 and 9 mean? What are the colors in Figure 6 (b) and (d) mean? What can be found from Figure 6? Also, the devices on the ping-pong racquet may mislead the reader because the paper proposed the devices for wristband, not for the racquet. Additionally, the acronyms should follow the full name, e.g., Physical activity recognition (PAR).

Reviewer 2 Report

The paper under review presents a method and system for detection of racquet sports in real time environment. The work though uses existing off-the-shelf computer vision components but the application is interesting. The paper is well-written with adequate experimental evaluation. The reviewer however could not verify the hardware proposed in the method.

Few minor explanations are needed in the manuscript. In the pre-processing step, signals of different modalities are fused to obtain the angle. It is, however, not clear from the text that how these signals are fused? and how what angle the context if referring to? The authors should provide more details on this.

The detection accuracy of Tennis stroke and drive actions are particularly low. It would be interesting to know the causes of this low detection rate. Similar is the case with the Badminton drive.

In the experimental evaluation, the results of the presented system are compared with existing similar vision based systems. The reviewer suggests that the detection accuracy of the system should be compared with vision based state-of-art action detection algorithms, such as,

S. V. Mora and W. J. Knottenbelt, "Deep Learning for Domain-Specific Action Recognition in Tennis," 2017 IEEE Conference on Computer Vision and Pattern Recognition Workshops (CVPRW), Honolulu, HI, 2017, pp. 170-178. Zhu, Guangyu, Changsheng Xu, Wen Gao, and Qingming Huang. "Action recognition in broadcast tennis video using optical flow and support vector machine." In European Conference on Computer Vision, pp. 89-98. Springer, Berlin, Heidelberg, 2006. Skublewska-Paszkowska, Maria, Edyta Lukasik, Bartłomiej Szydlowski, Jakub Smolka, and Pawel Powroznik. "Recognition of Tennis Shots Using Convolutional Neural Networks Based on Three-Dimensional Data." In International Conference on Man–Machine Interactions, pp. 146-155. Springer, Cham, 2019. Ó Conaire, Ciarán, Damien Connaghan, Philip Kelly, Noel E. O'Connor, Mark Gaffney, and John Buckley. "Combining inertial and visual sensing for human action recognition in tennis." In Proceedings of the first ACM international workshop on Analysis and retrieval of tracked events and motion in imagery streams, pp. 51-56. ACM, 2010. Zhu, Guangyu, Changsheng Xu, Qingming Huang, and Wen Gao. "Action recognition in broadcast tennis video." In 18th International Conference on Pattern Recognition (ICPR'06), vol. 1, pp. 251-254. IEEE, 2006. Vainstein, Jonathan, José F. Manera, Pablo Negri, Claudio Delrieux, and Ana Maguitman. "Modeling video activity with dynamic phrases and its application to action recognition in tennis videos." In Iberoamerican Congress on Pattern Recognition, pp. 909-916. Springer, Cham, 2014.

The term PAR in Line 31 is used in text without definition. Though its used as keyword but defining it in the text would be helpful for the reader.

Line 173: The sentence 'DBSCAN algorithm divides data with...' should be 'The DBSCAN algorithm divides data with...'

Fig. 9 should be properly commented/explained in the text.

The research presented in the manuscript is not concluded formally, a separate section with conclusions might be helpful.

Reviewer 3 Report

The paper describes a model and implementation for recognition of different movements during racquet sports baed on Inertial sensor signals. The model is based on rotation-invariant features and clustering. The results shows a high accuracy in the recognition when testing.

I liked the structure of the paper. The model is well-described and the results presented clearly. Section 2 is clearly described in a way that would make this study replicable by others. 

However, I cannot recommend it for acceptance because it lacks evaluation to confirm that the design is appropriate for the intended use case. It is not clear how the data was separated into train and evaluation sets which makes it difficult to assess the validity of the results. For instance, if one window is in the training set and the next window is in the testing set it is expected to have a high accuracy due to the overlapping of these windows. 

Moreover, one claim of the paper is that the system is suitable for real environments and fast calculations but there are no results to support this. I would have liked the authors to provide results on the time for a prediction to be done, communications lags, etc. Also, I would have liked an evaluation or discussion of the degradation of the sensor in the racquet due to use and impact of the ball.

Finally, there was no exploration of related work within the specific use case of racquet sports recognition, which leaves the reader confused about the actual contributions of this paper. 

see for example

Y. Wang, M. Chen, X. Wang, R. H. M. Chan and W. J. Li, "IoT for Next-Generation Racket Sports Training," in IEEE Internet of Things Journal, vol. 5, no. 6, pp. 4558-4566, Dec. 2018.
doi: 10.1109/JIOT.2018.283734

And related references (both cited and citing)